# Deciphering the Molecular Machinery—Influence of sE-Cadherin on Tumorigenic Traits of Prostate Cancer Cells

**DOI:** 10.3390/biology10101007

**Published:** 2021-10-07

**Authors:** Igor Tsaur, Anita Thomas, Eva Juengel, Sebastian Maxeiner, Timothy Grein, Quynh Chi Le, Veronika Muschta, Jochen Rutz, Felix K. H. Chun, Roman A. Blaheta

**Affiliations:** 1Department of Urology and Pediatric Urology, University Medicine Mainz, 55131 Mainz, Germany; anita.thomas@unimedizin-mainz.de (A.T.); eva.juengel@unimedizin-mainz.de (E.J.); 2Department of Urology, University Hospital Frankfurt, Goethe University, 60590 Frankfurt, Germany; sebastian.maxeiner@kgu.de (S.M.); timo.grein@hotmail.de (T.G.); quynhchi-le@hotmail.de (Q.C.L.); veronika.muschta@googlemail.com (V.M.); jochen.rutz@kgu.de (J.R.); felix.chun@kgu.de (F.K.H.C.); roman.blaheta@kgu.de (R.A.B.)

**Keywords:** prostate cancer, sE-cadherin, cell growth, migration, metastasis

## Abstract

**Simple Summary:**

Despite recent advances in the therapeutic management of metastasized prostate cancer, disease progression is still inevitable, with often fatal outcomes. Elucidating molecular mechanisms crucial to cancer development and progression is therefore necessary to find ways to interfere in metastatic processes and ultimately improve prognosis. Since soluble (s)E-cadherin is elevated in the serum of patients with prostate cancer, we investigated its influence on prostate cancer cell behavior in vitro. Exposure to sE-cadherin increased the systemic spread of the cells. Thus, targeting sE-cadherin might be a novel and innovative concept to treat advanced PCa.

**Abstract:**

The serum level of soluble (s)E-cadherin is elevated in several malignancies, including prostate cancer (PCa). This study was designed to investigate the effects of sE-cadherin on the behavior of PCa cells in vitro, with the aim of identifying a potential therapeutic target. Growth as well as adhesive and motile behavior were evaluated in PC3, DU-145, and LNCaP cells. Flow cytometry was used to assess cell cycle phases and the surface expression of CD44 variants as well as α and β integrins. Confocal microscopy was utilized to visualize the distribution of CD44 variants within the cells. Western blot was applied to investigate expression of α3 and β1 integrins as well as cytoskeletal and adhesion proteins. Cell growth was significantly inhibited after exposure to 5 µg/mL sE-cadherin and was accompanied by a G0/G1-phase arrest. Adhesion of cells to collagen and fibronectin was mitigated, while motility was augmented. CD44v4, v5, and v7 expression was elevated while α3 and β1 integrins were attenuated. Blocking integrin α3 reduced cell growth and adhesion to collagen but increased motility. sE-cadherin therefore appears to foster invasive tumor cell behavior, and targeting it might serve as a novel and innovative concept to treat advanced PCa.

## 1. Introduction

Despite the widespread use of prostate-specific antigen (PSA)-based screening together with steadily evolving molecular imaging and theranostics, morbidity and mortality due to prostate cancer (PCa) remain a significant burden to the male population. A contemporary analysis using the GLOBOCAN 2020 database estimated PCa to be the second leading malignancy and the fifth most common cause of cancer-related death in men worldwide, numbering 1,414,259 and 375,304 cases, respectively [1]. At the same time, global incidence and mortality rates of PCa based on the GLOBOCAN 2018 database have stabilized in most countries over the previous 5 years [2]. In the European Union, 67,100 deaths and an age-standardized mortality rate of 9.44 have been predicted for 2021, with the latter continuously and slowly decreasing after having peaked around the turn of the millennium [3]. Advances in diagnosis and surgical, radiological, and medical treatment have considerably contributed to improved outcome [3]. While localized PCa can be curatively managed by radiotherapy or radical prostatectomy, metastatic disease most often results in mortality. Recent introduction of efficacious systemic protocols to treat metastatic disease significantly prolongs survival. Docetaxel, androgen receptor blockers apalutamide and enzalutamide, as well as the CYP17 inhibitor abiraterone acetate, each combined with androgen deprivation therapy (ADT), have been shown to confer survival advantage in hormone-sensitive metastasized disease [4]. Advanced castration-resistant PCa clinical disease management has been improved by employing Poly (ADP-ribose) polymerase (PARP) inhibitors in men with BRCA1 or BRCA2 mutations together with radioligand therapy with Lutetium-177 (177Lu)-PSMA-617 [5].

Even with all these advances, progressive metastasized disease still cannot be stopped. Docetaxel treatment successfully delayed clinical or biochemical progression for only 19.4 months in the CHAARTED trial investigating upfront chemohormonal treatment vs. ADT in metastasized hormone-sensitive PCa. This underscores the unmet need to develop other potent systemic protocols [6]. The key to identifying possible therapeutic targets is elucidating molecular mechanisms crucial for tumorigenesis and cancer progression.

We have previously demonstrated that soluble (s) E-cadherin is overexpressed in the serum of patients with PCa and renal cell carcinoma (RCC) [7,8]. Thus, sE-cadherin, the ectodomain of the transmembrane adhesion protein and epithelial marker E-cadherin, offers itself as a promising target in PCa. sE-cadherin is cleaved from the membrane after proteolysis of full-length E-cadherin, provoking a disruption of adherens junctions and reducing cell aggregation capacity. As a paracrine/autocrine signaling molecule, sE-cadherin has been found to foster invasion and metastasis of various types of cancer, such as breast, ovarian, and lung cancer [9,10,11]. However, the relationship between sE-cadherin and PCa tumorigenesis and progression has as yet not been clearly delineated. The relation does seem evident though, since the serum level of sE-cadherin outperformed PSA in detecting tumors with a Gleason score of at least 7, and those with a Gleason score upgrade at radical prostatectomy as compared to a prostate biopsy. Moreover, treating RCC cells with sE-cadherin mitigates cell growth and adhesion but fosters migrative capacity, pointing to a function in tumor progression [7,8]. In the current project, we aimed to investigate how sE-cadherin affects the behavior of PCa cells, with the goal of identifying a potential therapeutic target to counteract tumor progression.

## 2. Materials and Methods

### 2.1. Cells Lines

Human, castration-resistant prostate tumor cell lines PC3, DU-145, and castration-sensitive LNCaP cells were purchased from DSMZ (Braunschweig, Germany). The prostate epithelial cell lines BPH-1 (from benign prostate hyperplasia) and PNT-1 (from normal prostate epithelium) were obtained from DSMZ and ATCC (Manassas, VA, USA). All cells were grown and subcultured in RPMI 1640 medium (Gibco/Invitrogen, Karlsruhe, Germany) augmented with 10% fetal calf serum (FCS), 2% HEPES (2-[4-(2-hydroxyethyl)piperazin-1-yl]ethanesulfonic acid) buffer (1 M, pH 7.4), 2% glutamine, 1% penicillin/streptomycin at 37 °C in a humidified, 5% CO_2_ incubator.

### 2.2. Cell Growth

Cell growth was assessed using the 3-(4,5-dimethylthiazol-2-yl)-2,5-diphenyltetrazolium bromide (MTT) dye reduction assay (Roche Diagnostics, Penzberg, Germany). PCa cells (100 µL, 1 × 10^4^ cells/mL) were seeded onto 96-well tissue culture plates and then treated with 0.5, 1, or 5 µg/mL sE-Cadherin (human, recombinant; CellSystems, Troisdorf, Germany). Controls were incubated without sE-cadherin. Following either 24, 48, or 72 h cell incubation, MTT (0.5 mg/mL) was added and allowed to react for 4 h. Cells were then lysed in a buffer containing 10% SDS in 0.01 M HCl. The plates were incubated overnight at 37 °C, 5% CO2. Absorbance at 550 nm was determined for each well using a microplate ELISA reader. Results were expressed as mean cell number after subtracting the background absorbance from the cell culture medium alone. The cell number after 24 h incubation was then set to 100%.

### 2.3. Cell Cycling

Cell cycle analysis was carried out on synchronized cell cultures. Tumor cells were synchronized at the G1-S boundary with aphidicolin (Sigma-Aldrich, Taufkirchen, Germany; 1 μg/mL) 24 h before starting cell cycle analysis and subsequently resuspended in fresh (aphidicolin-free) medium for 2 h. Cell populations were then treated with 5 µg/mL sE-cadherin (controls were not treated) for 24 h, stained with propidium iodide using a Cycle TEST PLUS DNA Reagent Kit (BD Biosciences, Heidelberg, Germany), and then subjected to flow cytometry (FACScalibur flow cytometer, BD Biosciences, Heidelberg, Germany). Ten thousand events were collected from each sample, and data were acquired using Cell-Quest software. ModFit software (BD Biosciences, Heidelberg, Germany) was used to assess cell cycle distribution. The number of gated cells in G0/G1-, G2/M-, or S-phase was presented as % of total cells.

### 2.4. Adhesion to Matrix Proteins

Six-well plates were coated with collagen G (extracted from calfskin, consisting of 90% collagen type I and 10% collagen type III (Seromed, Istanbul, Turkey) diluted to 400 µg/mL) or fibronectin (derived from human plasma; diluted to 50 µg/mL; BD Biosciences, Heidelberg, Germany) overnight. Plastic dishes served as the background control. Plates were washed with 1% bovine serum albumin (BSA) in phosphate buffered saline (PBS) to block nonspecific cell adhesion. Then, 0.5 × 10^6^ tumor cells, treated with culture medium alone or pre-treated with sE-cadherin at 0.5, 1, or 5 µg/mL, were added to each well for 60 min. Subsequently, non-adherent cells were fixed with 1% glutaraldehyde and counted microscopically. The mean cellular adhesion rate in each single experiment, defined by the number of cells adhering to the coated wells minus the number of cells adhering to the non-coated wells (background), was calculated from five different observation fields (5 × 0.25 mm^2^).

### 2.5. Chemotaxis

Serum-induced chemotactic movement was examined using six-well Transwell chambers (Greiner, Frickenhausen, Germany) with 8-µm pores. PC3 and DU145 cells were pre-treated with sE-cadherin (0.5, 5 µg/mL) for 4 h. Controls remained untreated. 0.5 × 10^6^ PC3 or DU145 cells per mL (LNCaP did not migrate and was therefore not included in the chemotaxis assay) were then placed in the upper chamber in serum-free medium. The lower chamber contained 10% serum. After 20 h incubation, the upper surface of the Transwell membrane was gently wiped with a cotton swab to remove non-migrating cells. Cells that had moved to the lower surface of the membrane were stained using hematoxylin and counted microscopically. The mean chemotaxis rate in each single experiment was calculated from five different observation fields (5 × 0.25 mm^2^).

### 2.6. Scratch Wound Assay

The scratch wound assay was used to examine horizontal migration of DU145 and PC3 cells in the presence of sE-cadherin. Tumor cells were incubated with 5 µg/mL sE-cadherin at 37 °C, 5% CO_2_ for 4 h and then seeded onto 96-well ImageLock plates (Sartorius, Goettingen, Germany) previously coated with 400 µg/mL collagen at 4 °C for 48 h (100 µL cell suspension, 5 × 10^5^ cells/mL). 24 h after plating out the cells, a defined scratch of about 700 µm was made with a IncuCyte^®^ WoundMaker (Sartorius, Göttingen, Germany). Detached cells were removed by washing with PBS with Ca^2+^ and Mg^2+^. Controls received cell culture medium without sE-cadherin. Plates were incubated in an IncuCyte^®^ Zoom (Sartorius, Göttingen, Germany) at 37 °C, 5% CO_2_ and photographed every 90 min for 15 h. Each experiment was done in triplicate. Relative wound density was calculated by the software WimScratch (Onimagin Technologies SCA, Córdoba, Spain).

### 2.7. CD44 Expression

Conjugating the CD44 variants 44v4, v5, and v7 antibodies was carried out with the Lightning-Link Allophycocyanin (APC) Conjugation Kit (eBioscience, ThermoFisher, Darmstadt, Germany). After detachment and washing, DU145, PC3, or LNCaP cells with blocking solution (PBS, 0.5% BSA) were then incubated for 1 h at 4 °C with 2.5 µL APC-conjugated monoclonal antibody directed against the following CD44 variants: anti-CD44v4 (clone VFF-11), anti-CD44v5 (clone VFF-8), and anti-CD44v7 (clone VFF-9; all: Bio-Rad, Feldkirchen, Germany). A total of 5 µL APC mouse IgG1, K (clone P3.6.2.8.1; ThermoFisher, Dreieich, Germany) served as the control isotype. CD44 expression was then assessed employing a FACScan (BD Biosciences, Heidelberg, Germany; FL4-H (log) channel histogram analysis; 1 × 10^4^ cells per scan) and expressed as mean fluorescence units (MFU). Alteration of CD44 expression by sE-cadherin (0.5, 1, 5 µg/mL) was evaluated in PC3 cells. Figures depict MFU data related to the controls set to 100%.

### 2.8. Confocal Laser Scanning Microscopy (CLSM)

To analyze CD44v4, v5, and v7 distribution in PC3, tumor cells (treated with 5 µg/mL sE-cadherin for 72h versus non-treated) were transferred to 8-well polystyrene culture slides (Falcon^®^; Merck Millipore, Darmstadt, Germany). Cell cultures were then washed and fixed in cold (−20 °C) methanol/acetone (60/40 *v*/*v*). Subsequently, cells were incubated for 60 min with APC-conjugated anti-CD44v4, v5, or v7 monoclonal antibodies. To prevent photobleaching of the fluorescent dye, cells were then embedded in an antifade reagent/mounting medium mixture including DAPI (VECTASHIELD, Antifade Mounting Media, Biozol, Eching, Germany). Cells were then viewed using a confocal laser scanning microscope (LSM 10; Zeiss, Jena, Germany) with a Plan-Neofluar ×63/1.3 oil immersion objective.

### 2.9. Integrin Surface Expression

Integrin surface expression was evaluated on PC3 treated with 5 µg/mL sE-cadherin and controls (untreated). Tumor cells were washed in blocking solution (PBS, 0.5% BSA) and then incubated for 60 min at 4 °C with phycoerythrin (PE)-conjugated monoclonal antibodies directed against the following integrin subtypes: anti-α1 (IgG1; clone SR84, dilution 1:1000), anti-α2 (IgG2a; clone 12F1-H6, dilution 1:250), anti-α3 (IgG1; clone C3II.1, dilution 1:1000), anti-α4 (IgG1; clone 9F10, dilution 1:200), anti-α5 (IgG1; clone IIA1, dilution 1:5000), anti-α6 (IgG2a; clone GoH3, dilution 1:200), anti-β1 (IgG1; clone MAR4, dilution 1:2500), or anti-β4 (IgG2a; clone 439-9B, dilution 1:250; all BD Biosciences, Heidelberg, Germany). Integrin surface expression was then measured using FACScan (BD Biosciences; FL-2H (log) channel histogram analysis; 1 × 10^4^ cells per scan) and expressed as mean fluorescence units (MFU). A mouse IgG1-PE (MOPC-21) or IgG2a-PE (G155-178; all: BD Biosciences) was used as an isotype control.

### 2.10. Western Blot Analysis

To evaluate the integrin proteins in PC3 cells exposed to sE-cadherin (versus untreated controls), tumor cell lysates were applied to a 7% polyacrylamide gel and electrophoresed for 90 min at 100 V. The protein was then transferred to nitrocellulose membranes (1 h, 100 V). After blocking with non-fat dry milk for 1 h, the membranes were incubated overnight with monoclonal antibodies against integrin α3 and integrin β1 (listed above (unconjugated)). Additionally, integrin-related signaling was investigated using anti-integrin-linked kinase (ILK; clone 3), anti-focal adhesion kinase (FAK; clone 77, and anti-phospho-specific focal adhesion kinase (FAK; pY397; clone 18) antibodies (all: BD Biosciences). Anti-phospho Akt (pAkt, pS472/pS473, clone 104A282; BD Biosciences) was used to investigate growth-related signaling. Cytoskeletal and focal adhesion proteins were analyzed as well by anti-talin 1 (rabbit IgG, clone C45F1), anti-α-actinin (polyclonal), anti-ezrin (polyclonal), anti-cytokeratin 7 (R458), anti-cytokeratin 8/18 (mouse IgG1, clone C51), anti-cytokeratin 19 (mouse IgG1, clone BA17; all: Cell Signaling, Leiden, the Netherlands), and anti-F-actin (mouse IgM, clone NH3; Gibco/Invitrogen, Karlsruhe, Germany).

Horseradish peroxidase (HRP)-conjugated goat anti-mouse IgG or IgM (Upstate Biotechnology, Lake Placid, NY, USA; dilution 1:5000) served as the secondary antibody. Membranes were briefly incubated with enhanced chemiluminescence (ECL) detection reagent (ECLTM, Amersham/GE Healthcare, München, Germany) to visualize the proteins and then analyzed by the Fusion FX7 system (Peqlab, Erlangen, Germany). β-actin (1:1000; Sigma, Taufenkirchen, Germany) served as the internal control.

### 2.11. Blocking Studies

PC3 cells were incubated for 60 min with 10 μg/mL function-blocking anti-integrin α3 (clone P1B5) or anti-integrin β1 (clone 6SG; all: Merck Millipore, Burlington, MA, USA). Controls were incubated with cell culture medium alone. Subsequently, tumor cell adhesion to immobilized collagen, fibronectin as well as chemotaxis were evaluated as described above. Cell growth of PC3 treated with function-blocking anti-integrin α3 or anti-integrin β1 were also analyzed by the MTT test as described. To explore the specificity of sE-cadherin, sE-cadherin-neutralizing antibody was added to PC3 or DU145 tumor cell cultures (Gibco/Invitrogen, Karlsruhe, Germany; clone HECD-1; 15 µg/mL) for 60 min. Cells were then incubated with sE-cadherin (versus sE-cadherin free and sE-cadherin containing medium without HECD-1) and then subjected to the MTT and chemotaxis assay.

### 2.12. Statistics

All experiments were performed 3–6 times, and statistical significance was determined with the Wilcoxon−Mann−Whitney U-test or Student’s *t*-test. Differences were considered statistically significant at *p* < 0.05.

## 3. Results

### 3.1. sE-Cadherin Blocks Tumor Growth

In androgen-resistant PC3 and DU145, sE-cadherin applied at 5 µg/mL significantly suppressed tumor growth. Growth of the androgen-sensitive cell line LNCaP was already diminished independent of the sE-cadherin concentration used (Figure 1A). Cell cycle analysis revealed an enhanced number of G0/G1 phase cells (all cell lines) along with a diminished number of S-phase (all cell lines) and G2/M-phase cells (DU145, PC3) (Figure 1B). We also evaluated whether sE-cadherin acts on non-cancerous epithelial cells using BPH-1 and PNT-2 cells. An amount of 5 µg/mL moderately suppressed growth of BPH-1 cells, derived from benign prostate hyperplasia, but did not influence growth of PNT-2 cells derived from normal prostate epithelium (Figure 1C).

### 3.2. sE-Cadherin Downregulates Cell Adhesion but Upregulates Cell Migration

Adhesion of PC3, DU145, or LNCaP cells to immobilized fibronectin was inhibited in the presence of 0.5, 1, or 5 µg/mL sE-cadherin. The same concentrations effectively prevented binding of DU145 to collagen, whereas 0.5 µg/mL sE-cadherin did not significantly block the interaction of PC3 and LNCaP to collagen (Figure 2A). 5 µg/mL sE-cadherin elevated PC3 chemotaxis, and both 0.5 and 5 µg/mL sE-cadherin increased the amount of chemotactic DU145 cells (Figure 2B). Migration of BPH-1 was slightly elevated in the presence of 5 µg/mL sE-cadherin, whereas migration of PNT-2 was not altered by sE-cadherin compared to controls (Figure 2C).

Results from the vertical chemotaxis assay were verified by the horizontal wound scratch assay. Wound closure occurred more rapidly when PC3 or DU145 cells were pretreated with sE-cadherin (5 µg/mL), compared to the untreated controls (Figure 3). LNCaP did not show any motile activity and were, therefore, not subjected to the chemotaxis studies.

### 3.3. sE-Cadherin Alters CD44v Expression

The CD44 variants v4, v5, and v7 were all expressed on PC3, DU145, and LNCaP cells (Figure 4A, Appendix A). Further studies, therefore, concentrated on PC3 cells which were incubated with sE-cadherin for different time periods. Fluorescence analysis was done after a 24, 48, and 72 h sE-cadherin incubation. Compared to the untreated controls, CD44v4 was already enhanced after 24 h and further increased after 48 and 72 h in the presence of 5 µg/mL sE-cadherin (Figure 4B). In contrast, CD44v5 was not altered after 24 h incubation, but a transient increase in CD44v5 was noted after 48 h with 0.5 or 1 µg/mL sE-cadherin. Strongest effects were seen following treatment with 5 µg/mL for 48 h, with a slight reduction of the CD44v5 expression level after 72 h (Figure 4B). sE-cadherin did not act on CD44v7 after 24 h but evoked distinct effects on this receptor at 5 µg/mL for 48h. Maximum effects of sE-cadherin on CD44v7 became evident after 72 h in the presence of 5 µg/mL sE-cadherin. At this time, 1 µg/mL sE-cadherin was also associated with elevated CD44v7 (Figure 4B). CD44v-specific antibodies were not available to permit cytoplasmic protein analysis by Western blotting. The distribution of CD44v was evaluated by CLSM instead. CD44v4, v5, and v7 were all located at the plasma membrane of PC3 cells in the control cell cultures. Following treatment with 5 µg/mL sE-cadherin, membranous accumulation was lost, and receptor proteins were homogenously located in the cytoplasm (Figure 4C).

### 3.4. sE-Cadherin Alters Integrin α3 and β1 Expression Level

The integrin subtypes α2, α3, α5, α6 (but not α4), β1, and β4 were all expressed on the PC3 cell surface (Figure 5A). Application of 5 µg/mL sE-cadherin induced a significant downregulation of α3 and β1, whereas further subtypes remained unaffected (Figure 5B). Western blotting pointed to a diminished expression of α3 and β1 proteins and reduced integrin-related signaling (FAK, pFAK, ILK). The level of pAkt was not significantly diminished under exposure to sE-cadherin, as compared to the control (Appendix A). Integrins α3 and β1 were then blocked to explore the relevance of these receptors for adhesion to collagen and fibronectin, chemotaxis, and growth regulation. Compared to control PC3 cells, blocking of α3 or β1 hindered cell attachment to collagen. Binding to fibronectin was diminished by β1 but not by α3 blockade. Interestingly, α3 level correlated negatively and that of β1 positively with the chemotactic activity of PC3 cells (Figure 5D). Cell growth of PC3 was reduced following treatment with either α3 or β1 function-associated antibodies (Figure 5E).

### 3.5. sE-Cadherin Influences Cytoskeletal and Focal Adhesion Proteins

In the subsequent experiment, integrin linkers and cytokeratins were evaluated in PC3 cells. Increased expression of talin, ezrin, α-actinin, and F-actin was induced by 0.5 and 5 µg/mL sE-cadherin (5 µg/mL > 0.5 µg/mL). In contrast, CK7, CK8/18, and CK19 became diminished both moderately and strongly following sE-cadherin exposure (Appendix A).

### 3.6. sE-Cadherin Blockade

Specific effects of sE-cadherin became apparent in the HECD-1 blocking experiment. sE-cadherin did not influence tumor growth (Figure 6A) or migration (Figure 6B) when the cells were pre-incubated with HECD-1 (each compared to the respective control).

## 4. Discussion

Identifying critical hubs of tumor molecular machinery is essential in developing efficacious cancer treatment options. Since the serum concentration of sE-cadherin has been reported to be upregulated in patients with several cancers including PCa [7,8,12,13], we aimed to explore the impact of sE-cadherin on PCa cell growth and invasive behavior.

Our in vitro data provide evidence that sE-cadherin suppresses PCa growth, presumably through accumulation of tumor cells in the G0/G1 cell cycle phase. For PCa, no data have been presented illuminating functional effects of sE-cadherin thus far. However, sE-cadherin has been shown to attenuate growth and proliferation of renal cell cancer cell lines [8]. Likewise, we show that sE-cadherin downregulated PCa growth activity. At the same time, chemotaxis and motility of PC3 and DU-145 cells were significantly enhanced under sE-cadherin. Since adhesion to collagen and fibronectin was reduced in parallel, it seems likely that sE-cadherin allows the tumor cells to detach from the extracellular matrix and begin motile crawling. Our observations concur with those of Johnson et al., who saw a reduction in pancreatic cancer cell aggregation in the presence of sE-cadherin, which was associated with increased tumor cell migration [14]. Accordingly, studies on renal cell cancer cells show a functional switch, induced by sE-cadherin, from sessility to aggressive dissemination [8]. Recent experiments carried out by Lee et al. with human embryonic kidney as well as colon, gastric, and ovarian cancer cell lines demonstrate sE-cadherin-triggered effects in a “cell- or context-dependent manner” [15]. These investigators found that sE-cadherin activates the Epidermal Growth Factor receptor (EGFr) and downstream Akt signaling. Importantly, they did not observe elevated cell proliferation. In contrast, we found Akt phosphorylation unaltered in response to sE-cadherin even though the migratory properties of both DU145 and PC3 cells were enhanced, which is in line with the observations made by Lee et al. [15]. A correlation between sE-cadherin level and metastatic dissemination has also been noted by others. Serum concentration of sE-cadherin increased during colorectal cancer metastasis in mice and in patients with late-stage cancer [12,16]. Furthermore, elevated sE-cadherin levels have been significantly associated with hepatic metastasis and muscle invasive bladder cancer [17,18]. Pre-operative plasma sE-cadherin levels served to identify bladder cancer patients with lymphatic metastasis and to predict disease progression [19]. We therefore conclude that sE-cadherin might facilitate metastatic cascade enhancing detachment of PCa cells from matrix proteins, which can result in systemic dissemination.

The underlying molecular mode of action of sE-cadherin has not been investigated yet. Our results point to an increased expression of CD44v4, v5, and v7 receptors on PC3 cells under sE-cadherin. However, the role of CD44 variants in tumor progression is discussed controversially. CD44v4, v5, and v7 have recently been observed to be involved in suppression of growth and proliferation in PC3 cells [20]. On the other hand, CD44v5 expression may correlate with poor outcome following radical prostatectomy, as reported by others [21]. At the protein level, CD44v7 is speculated to enhance PCa cell invasion [22]. We believe that CD44v4, v5, and v7 contribute to invasive PCa cell behavior. The time-dependency of CD44v expression should also be considered. Maximum effects of sE-cadherin on CD44v were initiated after 48 and 72 h but not after 24 h. Interestingly, translocation of CD44v from the surface membrane into the cytoplasm was observed. Whether this redistribution is transient and contributes to the behavioral switch of the tumor cells or is only an unspecific epiphenomenon is not yet clear. Therefore, the relevance of CD44 translocation in regard to invasive processes cannot be satisfactorily resolved. Based on the current literature and our results, we hypothesize that enhanced CD44v expression might be an important prerequisite for migration of tumor cells.

In regard to the integrin expression profile, the subtypes α3 and β1 were downregulated by sE-cadherin. Loss of both α3 and β1 was associated with diminished adhesion of PC3 cells to collagen, whereas β1 but not α3 was relevant for PC3 binding to fibronectin. Therefore, reduced tumor cell attachment to immobilized collagen and fibronectin under sE-cadherin might be associated with the suppression of α3 and/or β1. Expression of α3 also inversely correlated with PC3 chemotaxis, leading to the conclusion that diminishing α3 by means of sE-cadherin may, at least in part, be responsible for the elevated motile activity in sE-cadherin’s presence. Accordingly, depletion of α3 integrin increased cell migration of the PCa cell line DU145, as has recently been reported by Das et al. [23]. The role of integrins in tumor cell adhesion and migration is not yet completely understood. β1-blockade prevented chemotaxis and therefore stands in opposition to sE-cadherin, which elevated chemotaxis. A similar effect of β1 has been noted by others, where knockdown of integrin β1 significantly inhibited PC3 cancer cell migration and invasion in gain-of-function studies [24]. We postulate that down-regulation of α3 might be relevant for activating chemotaxis. The particular relevance of β1 in chemotactic processes requires further evaluation.

Loss of β1, as reported by Kwon et al. [25], results in attenuation of FAK phosphorylation and binding of PC3 cells to collagen. Hyder et al. provide evidence that pFAK may not influence PC3 chemotaxis but rather cell contact to matrix proteins [26]. Since FAK and pFAK (as well as ILK) were reduced by sE-cadherin in our experiments, we assume that this molecular mode of action is responsible for the diminished cellular adhesion capacity observed by exposure to sE-cadherin. When considering ILK/FAK function, it should also be recognized that ILK and FAK are also involved in PCa cell growth [27,28]. Probably, diminished ILK and FAK protein levels contribute to the reduced PCa cell viability in response to sE-cadherin as well. Whether ILK and FAK communicate with integrin α3 and/or β1 in this situation, or growth-related proteins are additionally involved, still remains to be elucidated.

The cytoskeleton-associated proteins talin, ezrin, α-actinin, and F-actin have all been reported to be upregulated by sE-cadherin [29]. Talin, in fact, is suggested to more accurately predict PCa lymph node metastasis than the Gleason score [30]. Association of CD44 with talin has been documented [31], which might explain why elevated talin is accompanied by elevated CD44v expression, as found in our study. Increased expression of ezrin correlated with lymph node metastasis of PCa [32] and is closely involved in the modulation of cytoskeletal dynamics, enhanced migration, and invasion [33]. α-actinin is another protein predictive of PCa aggressiveness [34]. In regard to F-actin, protein polymerization with well-structured actin bundles was proven necessary to speed up migration velocity [35]. Based on these findings, we hypothesize that the aforementioned cytoskeletal proteins may all serve as drivers of chemotactic movement of the tumor cells in our model system.

An inverse correlation between ezrin and CK8/18 has been observed in DU-145 cells and is ascribed to the acquisition of chemo-resistant features and a mesenchymal/metastatic phenotype [36]. Moreover, poorly differentiated PCa (Gleason score 8–9) showed a strong downregulation of CK 8, 18, and 19, compared to normal human prostate and benign prostatic hyperplasia [37], and studies on frozen tissue sections have revealed CK7 expression in nodular hyperplasia but not in prostate carcinoma [38]. Suppression of CK7, 8/18, and 19 by sE-cadherin could therefore be conducive to PCa dedifferentiation towards an invasive phenotype.

## 5. Conclusions

sE-cadherin reduces growth but enhances chemotaxis of PCa cells, whereby the enhanced chemotaxis is caused by loosening the tumor matrix contact. It is assumed that sE-cadherin promotes a behavioral switch of tumor cells to gain invasive properties. Several mechanisms might be responsible for these effects. These include upregulation of CD44v4, v5, and v7 and downregulation of the integrin subtypes α3 and β1, along with integrin-related signaling. In addition, sE-cadherin causes enhanced expression of the invasion-relevant cytoskeletal proteins talin, ezrin, F-actin, and α-actinin, and suppression of the epithelial markers CK7, 8/18 and 19. Therefore, targeting sE-cadherin might be a novel and innovative concept to treat advanced PCa.

## Figures and Tables

**Figure 1 biology-10-01007-f001:**
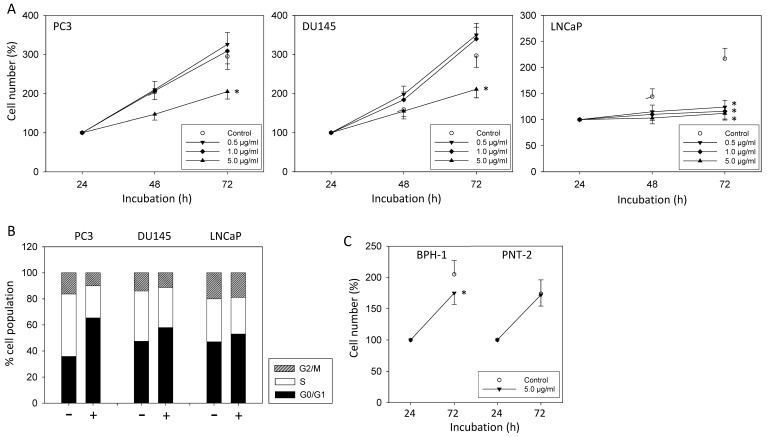
(**A**): Influence of sE-cadherin on cell growth of PC3, DU145, and LNCaP cells. Cell number evaluated after 24 (100%), 48, and 72 h by MTT assay. Error bars indicate standard deviation, * = *p* ≤ 0.05, *n* = 6. (**B**): Influence of 5 µg/mL sE-cadherin on proportionate G0/G1, S, and G2/M-phases of the cell cycle in PC3, DU145, and LNCaP cells over the course of 24 h (*n* = 3; * indicates significant difference to untreated controls). (**C**): Influence of sE-cadherin (5.0 µg/mL) on cell growth of BPH-1 and PNT-2 cells. Cell number evaluated after 24 (100%) and 72 h by MTT assay. Error bars indicate standard deviation, * = *p* ≤ 0.05, *n* = 3.

**Figure 2 biology-10-01007-f002:**
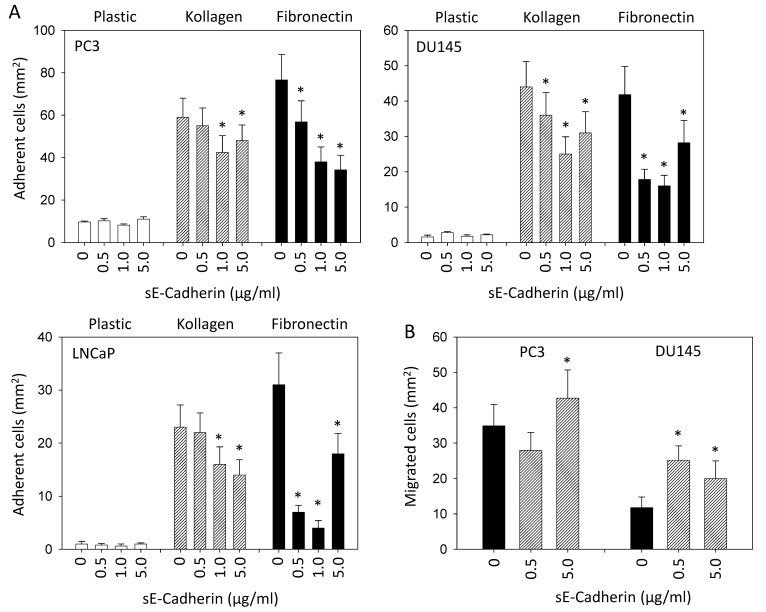
(**A**): Influence of sE-cadherin on adhesion of PC3, DU145, and LNCaP cells to plastic dishes, collagen- or fibronectin-coated plates. Five separate fields of 0.25 mm^2^ were counted at 200 × magnification (means ± SD, *n* = 6); * indicates significant difference to untreated controls. (**B**): Influence of sE-cadherin on chemotaxis of PC3 or DU145 towards a serum gradient after 24 h. 5 separate fields of 0.25 mm^2^ were counted at 200 × magnification (means ± SD, *n* = 6); * indicates significant difference to untreated controls. (**C**): Influence of sE-cadherin on chemotaxis of BPH-1 and PNT-2 cells towards a serum gradient after 24 h (means ± SD, *n* = 3); * indicates significant difference to untreated controls.

**Figure 3 biology-10-01007-f003:**
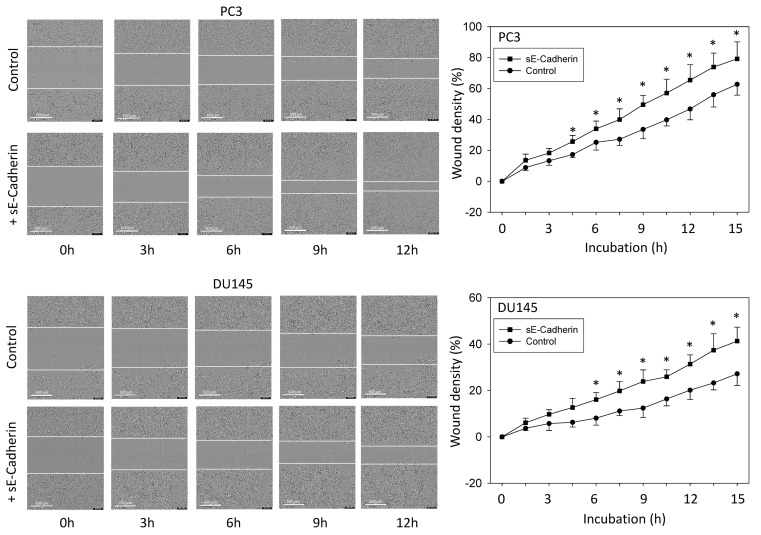
Wound closure analyzed after sE-cadherin exposure and compared to untreated controls. One representative figure and the mean wound closure of treated versus untreated PC3 or DU145 are shown (*n* = 3). Scale bar indicates 600 µm. * indicates significant difference to the untreated controls.

**Figure 4 biology-10-01007-f004:**
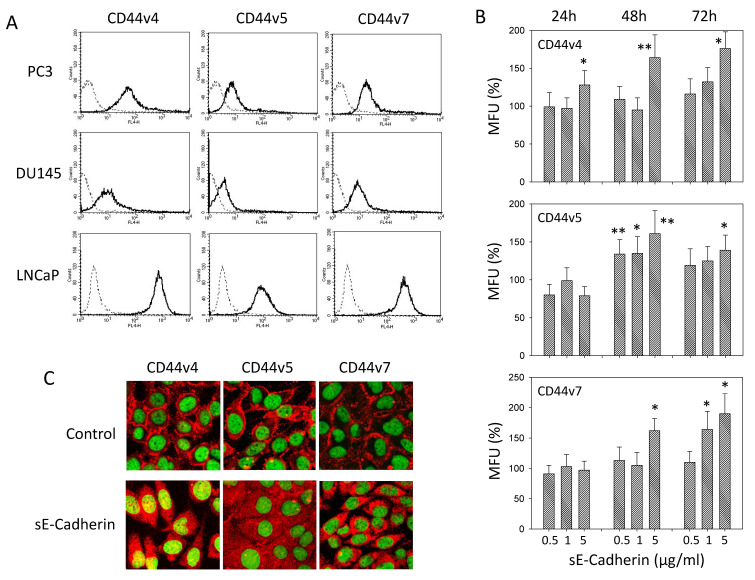
(**A**): CD44 variants v4, v5, and v7 on PC3, DU145, and LNCaP cells recorded by flow cytometry. Single representative of three separate experiments. Solid line: specific fluorescence; dashed line: isotype IgG1-APC. (**B**): CD44 variants v4, v5, and v7 on PC3 cells following sE-cadherin exposure for 24, 48 and 72 h. Means related to untreated controls (100%); MFU = mean fluorescence units; error bars indicate SD; * and ** = significant difference to corresponding control with *p* < 0.05 and *p* < 0.01, respectively, *n* = 3. (**C**): Distribution of CD44v4, v5, and v7 on PC3 cells after 72h sE-cadherin exposure. Pictures were taken by CLSM with a Plan-Neofluar × 63/1.3 oil immersion objective.

**Figure 5 biology-10-01007-f005:**
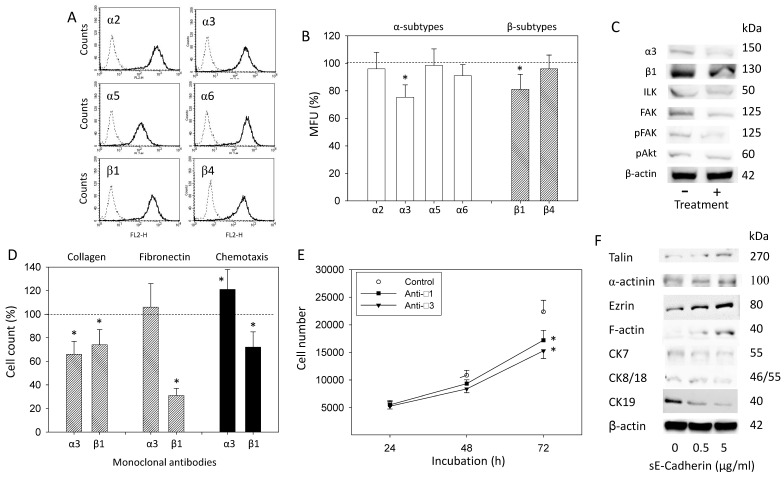
(**A**): Surface expression of α and β integrins on PC3 cells. Solid line: specific fluorescence, dashed line: IgG1-PE or IgG2a-PE. The abscissa shows the relative logarithmic distribution of the relative fluorescence intensity of α2, α3, α5, α6, β1, and β4. The ordinate shows cell number. 10,000 cells were counted. The figure is representative for *n* = 6. Scatter plots are shown in Appendix A. (**B**): Influence of sE-cadherin (5 µg/mL) on the integrin expression profile of PC3 cells. The untreated control is set to 100%. Values are means ± SD, *n* = 4; * indicates significant difference to controls. (**C**): Western blot of α and β integrins, ILK, FAK, pFAK, and pAkt in PC3 depending on the influence of sE-cadherin (5 µg/mL). Protein levels were measured 24 h after treatment. All bands are representative of *n* = 3. β-actin served as loading control and is representatively shown once. 50 µg were used per sample. (**D**): Adhesion to collagen and chemotaxis of PC3 cells after blockade of integrins α3 or β1. The untreated control is set to 100%. 5 separate fields of 0.25 mm^2^ were counted at 200× magnification (means ± SD, *n* = 3); * indicates significant difference to controls. (**E**): PC3 cell growth after blockade of integrins α3 or β1. Cell number evaluated after 24, 48, and 72 h by MTT assay. Error bars indicate standard deviation, * = *p* ≤ 0.05. (**F**): Western blot of cytoskeleton-related proteins in PC3 following sE-cadherin exposure (0.5, 5 µg/mL). Protein levels were measured after 24 h of treatment. All bands are representative of *n* = 3. β-actin served as loading control and is representatively shown once. 50 µg were used per sample.

**Figure 6 biology-10-01007-f006:**
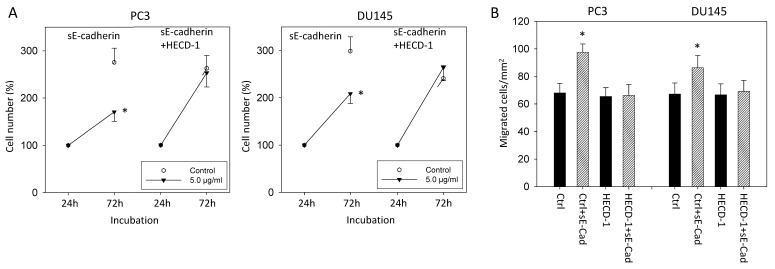
(**A**): Influence of sE-cadherin on cell growth of PC3 and DU145 cells treated with the sE-cadherin specific antibody HECD-1 (versus untreated). Cell number evaluated after 24 (100%) and 72 h by MTT assay. Error bars indicate standard deviation, * = *p* ≤ 0.05, *n* = 3. (**B**): Influence of sE-cadherin on chemotaxis of PC3 and DU145 cells treated with the sE-cadherin specific antibody HECD-1 (versus untreated). Endpoints after 24 h (means ± SD, *n* = 3); * indicates significant difference to controls. Ctrl: treatment with cell culture medium, HECD-1: treatment with the antibody alone, HECD-1+sE-Cad: treatment with HECD-1 and sE-cadherin (5 µg/mL).

## Data Availability

All data generated or analyzed during this study are included in this published article.

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
