# Peer review of "Deciphering the Molecular Machinery—Influence of sE-Cadherin on Tumorigenic Traits of Prostate Cancer Cells"

_biology, 2021, doi:10.3390/biology10101007_

Round 1

Reviewer 1 Report

The manuscript by Tsaur et al., investigate in vitro the effects of soluble E-cadherin on biological behavior of PCa cells and aim to discover potential therapeutic targets. Overall, this research is well written, it is a nicely designed and performed study and the content of this manuscript is of major interest. Notwithstanding the foregoing, the manuscript needs several elaborated modifications before acceptance for publication:

This paper would benefit from some closer proofreading. It is fraught with typographical mistakes. For instance, boldface is used in many parts of the text (e.g., lines 50-58) and should be avoided. moreover, a space should always be left between the number and the unit of measurement (see Materials and Methods, for instance 24h should be 24 h). Yet, in line 150 CO2 should be CO2. Please correct.

When cited in the text “figure x” needs a capital f. So, for instance, in line 237 “figure 1A” should be “Figure 1A”.

In figure 2, Kollagen (which I suppose is German) should be changed with Collagen.

Figures 3 (left) need a scale bar.

Line 330: “Western blotting pointed to a loss of α3 and β1 proteins”. Where? α3 is just slightly reduced and β1 is not lost.

Line 331: “The level of pAkt was slightly diminished…..” This is not true. The intensity of the band is the same in treated and untreated samples. This is clearly evident in Fig 5c and even more in the membranes reported in the supplementary material.

In Figure 5c you should indicate the kDa. The same in supplementary figures.

looking carefully at the supplementary figures, how can you say that the FAK and ILK are the correct bands?

WHAT Happens with Ezrin? The membrane in Fig 5f is not that some showed in Supplementary figures

In general, all these WB are “strange”. The bands you obtained are very weak and multiple no specific antigens are recognized by these antibodies. Furthermore, considering that you used loaded 50 ug of protein extract obtained from cell lines, ideally, in these conditions you should obtain very strong bands. Could the authors give me an explanation?

The discussion is very lengthy and partially repeat the results. Besides the authors focus partially their discission on results obtained in fig 5c and 5f that are noy very clear. As such, I suggest the author reduces this section to keep only the most important elements.

Line 402: “In fact, sE-cadherin also diminished the expression of the cell growth regulator pAkt”. This is not correct.

Author Response

Response letter Reviewer 1
The manuscript by Tsaur et al., investigate in vitro the effects of soluble E-cadherin on biological behavior of PCa cells and aim to discover potential therapeutic targets. Overall, this research is well written, it is a nicely designed and performed study and the content of this manuscript is of major interest. Notwithstanding the foregoing, the manuscript needs several elaborated modifications before acceptance for publication:

Comment 1: This paper would benefit from some closer proofreading. It is fraught with typographical mistakes. For instance, boldface is used in many parts of the text (e.g., lines 50-58) and should be avoided. moreover, a space should always be left between the number and the unit of measurement (see Materials and Methods, for instance 24h should be 24 h). Yet, in line 150 CO2 should be CO2. Please correct.

Our answer: We thank the author for this comment. We subjected the manuscript to proofreading by an English native speaker who changed the style and corrected typographical errors.

Comment 2: When cited in the text “figure x” needs a capital f. So, for instance, in line 237 “figure 1A” should be “Figure 1A”.

Our answer: We have corrected this in the whole manuscript.

Comment 3: In figure 2, Kollagen (which I suppose is German) should be changed with Collagen.

Our answer: Kollagen has been changed to Collagen in Figure 2 accordingly.

Comment 4: Figures 3 (left) need a scale bar.

Our answer: We apologize that the scale bar was not readable in the figure. This has been corrected. We also included in the figure legend: “Scale bar indicates 600 µm”.

Comment 5: Line 330: “Western blotting pointed to a loss of α3 and β1 proteins”. Where? α3 is just slightly reduced and β1 is not lost.

Our answer: It is correct that integrins were not lost but rather diminished by sE-cadherin. The respective sentence now reads (line 339): “Western blotting pointed to a diminished expression of α3 and β1 proteins”.

Comment 6: Line 331: “The level of pAkt was slightly diminished…..” This is not true. The intensity of the band is the same in treated and untreated samples. This is clearly evident in Fig 5c and even more in the membranes reported in the supplementary material.

Our answer: It was our initial intention to point to slight alterations of pAkt which, however, were not significant. We agree that this formulation is misleading. We have now clearly described the result (line 340): “The level of pAkt was not significantly diminished …”.

Comment 7: In Figure 5c you should indicate the kDa. The same in supplementary figures.

Our answer: Unfortunately, a preliminary version of the figure was shown in the submitted manuscript which did not show the kDa for all proteins. The correct, final, figure has now been included. We apologize for this mistake.

Comment 8: looking carefully at the supplementary figures, how can you say that the FAK and ILK are the correct bands?

Our answer: In all cases the protein ladder (molecular weight marker) was run in parallel. This is now shown in the supplements (FAK, ILK). The correct, final figure is now included in the manuscript, with ILK located at 50 kDa. Once again, we would like to apologize for our mistake.

Comment 9: WHAT Happens with Ezrin? The membrane in Fig 5f is not that some showed in Supplementary figures.

Our answer: The correct, final figure has now been included in the manuscript (please see also our answer to comment 7). The complete Ezrin blot is shown in the supplements.

Comment 10: In general, all these WB are “strange”. The bands you obtained are very weak and multiple no specific antigens are recognized by these antibodies. Furthermore, considering that you used loaded 50 ug of protein extract obtained from cell lines, ideally, in these conditions you should obtain very strong bands. Could the authors give me an explanation?

Our answer: Indeed, the load was 50 μg of total protein from the cell lysate. See line 367 However, the total protein content is not equivalent to the specific protein. Therefore, each band’s intensity following antibody staining depends on the amount of the specific protein. For example, integrin β1 was strongly enriched in the tumor cell extract, whereas the amount of α3 protein was low. Therefore, a strong β1 but a weak α3 band is visualized. Further conditions must also be considered; film exposure time, primary antibody dilution, and quality of the antibodies. Independent from antibody quality, extensive prolongation of the exposure time may lead to an increased background signal with unspecific staining. We did our best to optimize the protein staining, aware that we cannot present “ideal” protein bands.

Comment 11: The discussion is very lengthy and partially repeat the results. Besides the authors focus partially their discission on results obtained in fig 5c and 5f that are noy very clear. As such, I suggest the author reduces this section to keep only the most important elements.

Our answer: We have shortened the discussion to concentrate on the most important facts.

Comment 12: Line 402: “In fact, sE-cadherin also diminished the expression of the cell growth regulator pAkt”. This is not correct.

Our answer: We have removed this sentence (please also see our answer to comment 6).

Reviewer 2 Report

Clinical inclusion of prostate-specific antigen (PSA) lead to major advancement in the diagnosis and prognosis of prostate cancer. However, due to PSA’s lack of specificity, new biomarkers are needed to improve risk assessment and ensure optimal personalized therapy. E-cadherin is well-known for playing important roles in tumorigenesis as well as in tumor progression, invasion and metastasis. This protein exists in two forms: a membrane-tethered form and a soluble form.

Soluble E-cadherin ((s)E-cadherin) seems to be promising target for studies on future tools to be used in the management of prostate cancer. In the previous paper of the authors (see 10.1186/s13046-015-0161-6) it significantly correlated with the Gleason score in high risk cases and occurred to be independent prognostic factor. Furthermore, in other paper (s)E-cadherin proved to have greater specificity and accuracy than PSA (10.4172/2576-3962.1000115).

The production of (s)E-cadherin causes a reduction in cell aggregation capacity. As a paracrine/autocrine signaling molecule, it also activates or inhibits multiple signaling pathways and participates in the progression of various types of cancer, by promoting invasion and metastasis. 

As a consequence, authors focused on the new biomarker and successfully presented effects of sE-cadherin on biological behavior of PCa cells in in-vitro models, together with discovery of some potential therapeutic targets. The novelty and methodology of the experiments is indisputable. The experiments were conducted in different human cell lines, both castration-resistant and prostate epithelial cell lines. Most importantly, sE-Cadherin downregulated cell adhesion while upregulated cell migration. Then, authors proved that sE-cadherin dose not effect on non-cancerous epithelial cells and specific effects of sE-cadherin were HECD-1-dependent.

I would suggest to change the sentence in line 55: While localized PCa can be curatively managed with radiotherapy, radical prostatectomy, focal therapy and active surveillance into: While localized PCa can be curatively managed with radiotherapy or radical prostatectomy.

Author Response

Response letter Reviewer 2

Clinical inclusion of prostate-specific antigen (PSA) lead to major advancement in the diagnosis and prognosis of prostate cancer. However, due to PSA’s lack of specificity, new biomarkers are needed to improve risk assessment and ensure optimal personalized therapy. E-cadherin is well-known for playing important roles in tumorigenesis as well as in tumor progression, invasion and metastasis. This protein exists in two forms: a membrane-tethered form and a soluble form.

Soluble E-cadherin ((s)E-cadherin) seems to be promising target for studies on future tools to be used in the management of prostate cancer. In the previous paper of the authors (see 10.1186/s13046-015-0161-6) it significantly correlated with the Gleason score in high risk cases and occurred to be independent prognostic factor. Furthermore, in other paper (s)E-cadherin proved to have greater specificity and accuracy than PSA (10.4172/2576-3962.1000115).

The production of (s)E-cadherin causes a reduction in cell aggregation capacity. As a paracrine/autocrine signaling molecule, it also activates or inhibits multiple signaling pathways and participates in the progression of various types of cancer, by promoting invasion and metastasis.

As a consequence, authors focused on the new biomarker and successfully presented effects of sE-cadherin on biological behavior of PCa cells in in-vitro models, together with discovery of some potential therapeutic targets. The novelty and methodology of the experiments is indisputable. The experiments were conducted in different human cell lines, both castration-resistant and prostate epithelial cell lines. Most importantly, sE-Cadherin downregulated cell adhesion while upregulated cell migration. Then, authors proved that sE-cadherin dose not effect on non-cancerous epithelial cells and specific effects of sE-cadherin were HECD-1-dependent.

Comment 1: I would suggest to change the sentence in line 55: While localized PCa can be curatively managed with radiotherapy, radical prostatectomy, focal therapy and active surveillance into: While localized PCa can be curatively managed with radiotherapy or radical prostatectomy.

Our answer: We have changed the sentence in lines 53-55 accordingly. It now reads: “While localized PCa can be curatively managed by radiotherapy or radical prostatectomy, metastatic disease most often results in mortality.”.

Reviewer 3 Report

The manuscript by Tsaur et al. describes the effect of sE-cadhein on various prostate cancer cells. The authors use several methods including cell proliferation assay, flow cytometry, adhesion to matrix proteins, chemotaxis, scratch wound assay, expression of various proteins involved in cell migration/proliferation. It is concluded that sE-cadherin plays important in switching prostate cancer cells from being proliferative to being invasive. This is important finding as it puts up sE-cadherin as a novel target in prostate cancer.

The manuscript is mostly clearly written with occasional typographical errors. Some figures are of low quality and should be replace with ones in high resolution.

My major concern is however about the design. The control with BPH-1 and PNT-2 cells is not mentioned until the last section. It appears that the authors remembered to include that control at the last moment, after all the test had been done. As a result the manuscript lacks uniformity. Incorporating these control cells along all experiments would have possibly provided better insight into potential differences between cancer and non-cancer cells in regard to their response to sE-cadherin. It is also worth mentioning at this point the text reads "Migration of BPH-1 was also slightly diminished..." (line 374) whereas the figure 6B shows it to be elevated.

Some minor points:

  1. Please rephrase the sentence "sE-cadherin is cleaved from..." (line 75) as it is long and difficult to understand.
  2. Abiraterone MOA is mainly CYP17A1 inhibition rather than AR (line 59)
  3. MTT addition (line 107) is confusing. Is it added after 4 hours of initial timepoints (24 h, 48 h, 72 h) or added at the time specified and then allowed to react for 4 hours after which cells were lysed? Please clarify.
  4. Low concentrations (0.5 µg / mL and 1.0 µg / mL) increase proliferation in DU145 (line 235). Could authors comment a bit about that?
  5. CD44 variants are expressed on all PC3, DU145 and LNCap cells. Yet authors select PC3 cells for further testing. Was there any rationale behind it or was it random selection?
  6. Some typographical errors as mentioned before (eg. line 459 *be responsible, line 504 *our model system, fig.2 *Collagen, etc).
  7. Please consider minor style changes throughout the manuscript, eg. line 441 taken care of -> accounted for.

Author Response

Response letter Reviewer 3
The manuscript by Tsaur et al. describes the effect of sE-cadhein on various prostate cancer cells. The authors use several methods including cell proliferation assay, flow cytometry, adhesion to matrix proteins, chemotaxis, scratch wound assay, expression of various proteins involved in cell migration/proliferation. It is concluded that sE-cadherin plays important in switching prostate cancer cells from being proliferative to being invasive. This is important finding as it puts up sE-cadherin as a novel target in prostate cancer.

The manuscript is mostly clearly written with occasional typographical errors. Some figures are of low quality and should be replace with ones in high resolution.

Comment 1: My major concern is however about the design. The control with BPH-1 and PNT-2 cells is not mentioned until the last section. It appears that the authors remembered to include that control at the last moment, after all the test had been done. As a result the manuscript lacks uniformity. Incorporating these control cells along all experiments would have possibly provided better insight into potential differences between cancer and non-cancer cells in regard to their response to sE-cadherin. It is also worth mentioning at this point the text reads "Migration of BPH-1 was also slightly diminished..." (line 374) whereas the figure 6B shows it to be elevated.

Our answer: Initially, we weren’t sure whether to include the data of the “normal” epithelial cells. Although both BPH-1 and PNT-2 cells are defined as “normal” epithelial cells, they have been immortalized and are therefore characterized by distinct growth activity. To comply with the referee’s comment, data is now presented together with the growth and migration data of the tumor cell lines (figures 1+2). The manuscript text has been changed accordingly. Figure 6 now concentrates on the sE-cadherin blocking studies. The mistake in line 258 has been corrected.

Comment 2: Please rephrase the sentence "sE-cadherin is cleaved from..." (line 75) as it is long and difficult to understand.

Our answer: We modified this sentence in lines 73-77 to read: “sE-cadherin is cleaved from the membrane after proteolysis of full-length E-cadherin, provoking a disruption of adherens junctions and reducing cell aggregation capacity. As a paracrine/autocrine signaling molecule, sE-cadherin has been found to foster invasion and metastasis of various types of cancer, such as breast, ovarian and lung cancer ”.

Comment 3: Abiraterone MOA is mainly CYP17A1 inhibition rather than AR (line 59)

Our answer: To address this aspect, we modified the appropriate sentence in lines 56-59 which now reads: “Docetaxel, androgen receptor blockers apalutamide and enzalutamide as well as the CYP17 inhibitor abiraterone acetate, each combined with androgen deprivation therapy (ADT), have been shown to confer survival advantage in hormone-sensitive metastasized disease [4].”

Comment 4: MTT addition (line 107) is confusing. Is it added after 4 hours of initial timepoints (24 h, 48 h, 72 h) or added at the time specified and then allowed to react for 4 hours after which cells were lysed? Please clarify.

Our answer: MTT was added at the time specified, either after 24, 48h, or 72h and then allowed to react for 4 hours. To make this clearer, the respective methods part now reads (line 103): “Following either 24, 48, or 72 h cell incubation, MTT (0.5 mg/mL) was added and then allowed to react for 4 h”.

Comment 5: Low concentrations (0.5 µg / mL and 1.0 µg / mL) increase proliferation in DU145 (line 235). Could authors comment a bit about that?

Our answer: Indeed, there was a trend towards an increased cell growth rate in the presence of low sE-cadherin concentrations (at least with DU145 and PC3 cells). Due to this, we formulated the hypothesis of a functional switch of sE-cadherin, which may depend on its concentration. We started new experiments to confirm this, however still failed to demonstrate a strong biphasic mechanism of sE-cadherin. Therefore, we would prefer not to discuss this uncertain phenomenon in the present paper. 

Comment 6: CD44 variants are expressed on all PC3, DU145 and LNCap cells. Yet authors select PC3 cells for further testing. Was there any rationale behind it or was it random selection?

Our answer: It is correct that CD44 variants are expressed on all cell lines. LNCaP cells were excluded from further studies since they do not display migratory activity (Please note: Figures 2B and 3 are related to DU145 and PC3 cells only). PC3 was then randomly selected to proceed with the experiments. We are aware that the data presented in figure 5 do not allow assumption of the same molecular mechanisms for DU145 cells. Differences in molecular mechanisms are common in different cell lines from the same type of cancer. So as not to complicate the overall message, this aspect has not been dealt with in the present manuscript,

Comment 7: Some typographical errors as mentioned before (eg. line 459 *be responsible, line 504 *our model system, fig.2 *Collagen, etc). Please consider minor style changes throughout the manuscript, eg. line 441 taken care of -> accounted for.

Our answer: We subjected the manuscript to a proofreading by an English native speaker who changed the style and corrected typographical errors.

Reviewer 4 Report

This manuscript reports the study of the sE-cadherin system in series of common prostate cancer cell lines and non-malignant cell lines.  In general, the work appears sound and support the idea that targeting the sE-cadherin system might have a beneficial effect  on PCa.  For the most part the manuscript is well-written, although there are several places where the English is not clear.  The manuscript therefore needs to be edited by a native English speaker who is familiar with the work, but not associated with it.

The following points need clarification as well:

Line 268.  It is not clear just how many replicates were used because the authors give conflicting numbers in: “Five separate fields of 0.25 mm2 were counted at 200× magnification (means ± SD, n = 6).”

Line 362.  The same confusion reigns in: “5 separate fields of 0.25 mm2 were counted at 200× magnification (means ± SD, n = 3). ”

Line 421. “Although phosphorylation of Akt was not observed in our experiments…” I believe the authors mean that they did not measure the phosphorylation of Akt in this study.

Line 504.  I believe the authors meant to say “…chemotactic movement of the tumor cells in our model system.”

Author Response

Response letter Reviewer 4
This manuscript reports the study of the sE-cadherin system in series of common prostate cancer cell lines and non-malignant cell lines.  In general, the work appears sound and support the idea that targeting the sE-cadherin system might have a beneficial effect on PCa.  For the most part the manuscript is well-written, although there are several places where the English is not clear. 

Comment 1: The manuscript therefore needs to be edited by a native English speaker who is familiar with the work, but not associated with it.

Our answer: We have had an English native speaker, familiar with the subject, do editing directed towards clarity.

Comment 2: Line 268.  It is not clear just how many replicates were used because the authors give conflicting numbers in: “Five separate fields of 0.25 mm2 were counted at 200× magnification (means ± SD, n = 6).”

Comment 3: Line 362.  The same confusion reigns in: “5 separate fields of 0.25 mm2 were counted at 200× magnification (means ± SD, n = 3).”

Our answer: We apologize for this confusion. In each single adhesion and migration (chemotaxis) experiment, cells were counted at five different areas of the culture plates (adhesion experiment) or on the Transwell membrane (chemotaxis experiment), i.e. central area, up and down from the central area, left and right from the central area. Each area covered 0.25 mm2. The mean value from the five counts was then calculated. The experiments were done 6 times for adhesion and 3 times for migration, indicated in respective figures. To make this point clearer, we have included into methods (line 129): “The mean cellular adhesion rate in each single experiment,” ….. and (line 142) : “The mean chemotaxis rate in each single experiment …”.

Comment 4: Line 421. “Although phosphorylation of Akt was not observed in our experiments…” I believe the authors mean that they did not measure the phosphorylation of Akt in this study.

Our answer: We thank the reviewer for this comment. Lines 411-416 now read: “These investigators found that sE-cadherin activates the Epidermal Growth Factor receptor (EGFr) and downstream Akt signaling. Importantly, they did not observe elevated cell proliferation. In contrast, we found AKT phosphorylation unaltered in response to sE-cadherin though the migratory properties of both DU145 and PC3 cells were enhanced, which is in line with the observations made by Lee et al.

Comment 5: Line 504. I believe the authors meant to say “…chemotactic movement of the tumor cells in our model system.”

Our answer: Yes, thanks. We changed it accordingly. The sentence now reads (line 477): Based on these findings, we hypothesize that the aforementioned cytoskeletal proteins may all serve as drivers of chemotactic movement of the tumor cells in our model system”.

Round 2

Reviewer 1 Report

The authors have addressed all my comments/suggestions. I found their responses quite satisfactory and the revised version has been much improved. I now recommend the paper for publication in Biology.

Reviewer 3 Report

The authors addressed the raised issues. Thank you for your response. Please remember to use high quality figures in the final proof. It appears they are even worse now than in the original manuscript.